# Using Mind–Body Modalities via Telemedicine during the COVID-19 Crisis: Cases in the Republic of Korea

**DOI:** 10.3390/ijerph17124477

**Published:** 2020-06-22

**Authors:** Chan-Young Kwon, Hui-Yong Kwak, Jong Woo Kim

**Affiliations:** 1Department of Oriental Neuropsychiatry, Dong-eui University College of Korean Medicine, Busan 47227, Korea; beanalogue@naver.com; 2Department of Neuropsychiatry, Kyung Hee University Korean Medicine Hospital at Gangdong, Seoul 05278, Korea; polestar92@hanmail.net

**Keywords:** COVID-19, telemedicine, new-normal, mind–body intervention

## Abstract

The coronavirus disease 2019 (COVID-19) pandemic affected the world, and its deleterious effects on human domestic life, society, economics, and especially on human mental health are expected to continue. Mental health experts highlighted health issues this pandemic may cause, such as depression, anxiety, obsessive compulsive disorder, and post-traumatic stress disorder. Mind–body intervention, such as mindfulness meditation, has accumulated sufficient empirical evidence supporting the efficacy in improving human mental health states and the use for this purpose has been increasing. Notably, some of these interventions have already been tried in the form of telemedicine or eHealth. Korea, located adjacent to China, was exposed to COVID-19 from a relatively early stage, and today it is evaluated to have been successful in controlling this disease. “The COVID-19 telemedicine center of Korean medicine” has treated more than 20% of the confirmed COVID-19 patients in Korea with telemedicine since 9 March 2020. The center used telemedicine and mind–body modalities (including mindfulness meditation) to improve the mental health of patients diagnosed with COVID-19. In this paper, the telemedicine manual is introduced to provide insights into the development of mental health interventions for COVID-19 and other large-scale disasters in the upcoming new-normal era.

## 1. COVID-19, Telemedicine, and Mind–Body Interventions

The coronavirus disease 2019 (COVID-19) was first reported as “a cluster of cases of pneumonia” on 31 December 2019 at the Wuhan Municipal Health Commission in China. On 30 January 2020, World Health Organization (WHO) Director-General declared the novel coronavirus outbreak as a Public Health Emergency of International Concern (PHEIC) and eventually declared COVID-19 as a pandemic on 11 March [1]. The optimal treatment for infection of this coronavirus, which has the official name of severe acute respiratory syndrome coronavirus 2 (SARS-CoV-2), has not yet been established, and symptomatic treatment and supportive treatment are mainly performed [2]. Therefore, front-line health care providers and health authorities around the world had great difficulties during this pandemic. Antiviral agents approved for treatment of common influenza, such as favilavir, are being tried on COVID-19 patients, but the established clinical evidence is lacking, and clinical trials of the first vaccine against SARS-CoV-2 are unlikely to be conducted this year [3].

Korea, an adjacent country to China, was affected by COVID-19 from a relatively early stage, and today it is evaluated to have been successful in extinguishing COVID-19. As of 4 May 2020, the Ministry of Health and Welfare’s Central Disaster Management Headquarters and Central Preventive Measures Headquarters in Korea reported the number of confirmed patients was 8 and the cumulative confirmed number of patients was 10,801 since 15 February 2020 (Figure 1) [4].

Although strategies to cope with the physical health impact of COVID-19 were gradually explored, the impact of this pandemic on human families, society, economics, and especially on the mental health of individuals, are expected to continue [5,6,7]. First, the impact of the pandemic on mental health started with the ambiguity and anxiety that arose from the absence of a cure or vaccine for SARS-CoV-2 infection, and the rapid spread of infection brought panic to the public with limited quarantine resources [8]. Many people had to watch themselves or beloved family and friends suffer from the disease. Moreover, many people lost the freedom of everyday life due to social distancing and/or quarantine. Furthermore, some argue individuals must prepare for the “new-normal” [9]. In addition to the negative psychological effects of disease-related factors, the worldwide economic contraction caused by this epidemic is considered a major threat to the survival of the general public from an economic perspective [10].

One encouraging factor is the use of telemedicine during the pandemic [11]. Since COVID-19 makes it impossible for patients with various diseases to visit a clinic directly for face-to-face care due to concerns over the spread of the virus, several healthcare providers are looking for ways to use telemedicine instead [12]. Mental health management is one of the areas that is making rapid progress in the field of telemedicine. Mindfulness, a type of mind–body modality, may be defined as one being non-judgmentally aware in every moment. In the field of mental health care, the combination of telemedicine or e-Health has been tried along with mindfulness [13].

Korea has a dualized medical system called Western medicine and Korean medicine (KM) and “the COVID-19 telemedicine center of KM” has treated more than 20% of confirmed COVID-19 patients in Korea with telemedicine since 9 March (Figure 2). Initially, the telemedicine center was established by the Association of Korean Medicine (AKOM), a representative organization of all KM doctors established in the early 1950s, at Daegu Korean Medicine Hospital, Daegu, Korea, where the number of COVID-19 patients increased rapidly. After the outbreak was controlled in Daegu and Gyeongbuk, AKOM set up a second COVID-19 telemedicine center of KM in Seoul. The center uses herbal medicine (mainly *Qing-Fei-Pai-Du-Tang*) with established protocols based on multidisciplinary expert discussions and empirical evidence [14], and the center also used telemedicine in conjunction with mind–body modalities (including mindfulness meditation) to improve the mental health of COVID-19 patients in Korea. The aim of this paper is to introduce the KM doctor’s mental health instruction manual in telemedicine for COVID-19 and to provide insights into the development of mental-health interventions for COVID-19 patients and large-scale disasters in the upcoming “new-normal” era.

## 2. Korean Medicine (KM) Doctor’s Mental Health Instruction Manual in Telemedicine for COVID-19

### 2.1. Development of the Manual

To better manage the mental health of COVID-19 victims and survivors, the COVID-19 telemedicine center of KM requested specialists in Oriental neuropsychiatry to develop the KM doctor’s mental health instruction manual in telemedicine for COVID-19 in a written format. Specifically, the manual was developed by professors, professor Jong Woo Kim who is also the corresponding author of this article and Professor Sun-Yong Chung, at the Department of Oriental Neuropsychiatry at KyungHee University in Korea. Moreover, they practice the Hwa-byung and Stress Clinic at the KyungHee University Korean Medicine Hospital at Gangdong and have used mindfulness meditation in their clinical practice for over 10 years. Professor Kim, the primary author of this manual, is also the president of the Korean Society for Meditation [15]. As is the case in other countries, the COVID-19 pandemic caused an unprecedented crisis in Korea and there were no previously reported cases of using telemedicine for mental health during such catastrophes. In addition, due to the urgency of developing the manual there was not sufficient time to conduct a comprehensive systematic review of the literature regarding the development of such a manual and intervention for human subjects. Instead, the two professors and an assistant, Dr. Hui-Yong Kwak who is one of author of this article, developed this manual using empirical evidence obtained from not-systematic review methodology as well as clinical experiences from their clinical settings.

### 2.2. Objectives

This manual was intended to enable KM doctors to provide appropriate guidance and counseling for individuals who needed mental health care via a novel method of telemedicine. Specifically, it provided guidance on managing psychological problems, such as anxiety, depression, fear, and anger, and related physical symptoms such as pain, digestive problems, and insomnia. Throughout the manual, KM doctors categorized the potential psychological condition of individuals, and explained the symptoms that may happen in this stressful situation and guided potentially useful mind–body interventions.

### 2.3. Target Population

The target population of the manual was primarily confirmed COVID-19 patients. In addition, it could be applied to people who were self-contained in contact with the confirmed patient, the family and acquaintances of the confirmed patient, and the general public complaining of anxiety related to COVID-19.

### 2.4. Three Steps of History Taking

To identify the potential psychological condition of individuals, the following three steps were used. The majority of these questions are leading, since this manual was applied to all individuals receiving telemedicine services at this center and the number of staff (mostly volunteers) was limited. Therefore, questions were constructed to quickly and efficiently assess the need for mental health care and identify the presence of related symptoms (Table 1).

### 2.5. Guidance for Symptom Management

In the telemedicine of the KM center, KM doctors explained the mechanism of symptom occurrence that individuals complain about, and each recommended modality. All counseling required an explanation of the current situation and empathy for the individual (Table 2).

### 2.6. Care Algorithm for Individuals

This manual consists of care algorithms for individuals that included the symptom, education on each symptom, basic modalities, and individual mind–body modalities. Based on this care algorithm, three basic modalities including simple breathing, mindful breathing, and walking meditation were suggested twice a day for overall mental health improvement. For each symptom that an individual complained of, individual mind–body modalities were suggested to improve the symptom (Table 3).

### 2.7. Some Advice on Self-Management

A description of the mind–body modalities for symptom management is provided in Table 4. Individuals may achieve efficient self-management of their symptoms through YouTube videos in which detailed instructions for each modality are provided [16].

### 2.8. Frequently Asked Questions

KM doctors explained why immunity plays an important role in the prevention and recovery of virus infection, including SARS-CoV-2 infection, to the individuals [17]. In addition, the impact of mind health on immunity was also explained.


**Question 1.**
*Why is immunity important in the COVID-19 era?*


**Answer 1.** Currently, no cure or vaccine for the new coronavirus has been developed. Although no such treatments or vaccines have been developed, there are thousands of COVID-19 patients who are self-healing. This is because the most important weapon against viruses is the body’s immunity. Since SARS-CoV-2 inhibits cells involved in immunity when it enters the human body, it is best to maintain and regulate immune function before the infection.


**Question 2.**
*Does our mind or mood affect immunity?*


**Answer 2.** Depression or anxiety can decrease the activity of immune cells and increase the level of inflammation, thereby increasing the risk of preventing the body from responding appropriately to viral infections. Short-term stress causes the human body to produce more immune cells, but long-term stress rather causes an accumulation of inflammatory substances, disrupting the immune system homeostasis.


**Question 3.**
*Are there any recommended actions to improve immunity?*


**Answer 3.** Eating foods rich in antioxidants; getting enough sleep; exercising regularly; and it is important to get away from excessive psychological stress. Simple activities such as singing, exercising, foot bathing, or even watching comedy movies, can help boost the release of immune-related molecules.


**Question 4.**
*Are there any contraindications that can impair immunity?*


**Answer 4.** Eating junk food; breaks in life rhythms including sleep cycles; decreased physical activity; anxiety, repeated psychological stress, among others, have a detrimental effect on immunity. In particular, the sleep cycle plays a very important role in regulating the cycle of the immune system, and chronic sleep deprivation affects the balance of inflammatory cytokines, causing vulnerability to hypersensitivity reactions. In addition, excessive anxiety about the future and regret about the past can exacerbate negative emotions and physical symptoms, which can act as a burden on the immune system.


**Question 5.**
*Does meditation help improve immunity?*


**Answer 5.** As a result of measuring the inflammation level and activity of the immune system in various studies, it was reported that when meditating, the body’s inflammation level was lowered and the immune system activity was increased. Representatively, there are some studies of mindfulness meditation and loving-kindness meditation.


**Question 6.**
*What is the principle of relaxation?*


**Answer 6.** Relaxation can be applied as a simple means to reduce physical and mental tension. The scope of relaxation is very comprehensive, and the breathing method is also used as a preparation step before starting a full-scale meditation. In relaxation, focusing on body sensations is a key concept. The most effective way to stay in the ‘here and now’ is to focus on the body sensations.


**Question 7.**
*What is mindfulness?*


**Answer 7.** Mindfulness is ‘observing what is, as it is.’ It is to create a state of staying here after putting down a lot of thoughts that arise automatically, such as certain preconceptions and stereotypes. In mindfulness, one’s mind is not deceived by thoughts or other body sensations and can focus on a specific object or phenomenon itself. In fact, if you look at your body and mind excluding “interpretation” and ”prejudice” among others, you will notice that you are in a clearer mind and more comfortable body than before.


**Question 8.**
*What is loving-kindness meditation?*


**Answer 8.** Loving-kindness meditation is often called social meditation. Loving-kindness is the desire for people to be peaceful and happy, and compassion is the desire for people to escape from suffering. In a situation of disconnected and lonely alienation, you will be able to fill the natural energy and solidarity of human beings by having a warm heart and by practicing passing it on to others or to yourself.

## 3. Limitations of the Manual and Further Suggestions

With the introduction of this manual, we look forward to the widespread use of mind–body medicine, including mindfulness-based interventions, to improve mental health in other disaster areas. However, in order for this manual to be applied in other environments, some limitations must first be taken into account. First, simple mental health measures should be introduced to simplify the evaluation of individuals’ mental health in the telemedicine environment. Since the creation and implementation of our manual was conducted during a pandemic, and not for the purpose of a study, it was insufficient to consider it as a proper outcome indicator. Some indicators, such as the Beck Depression Inventory, the Beck Hopelessness Scale, the Hamilton Anxiety Rating Scale, and the Pittsburg Sleep Quality Index may be considered [18], but considering the nature of telemedicine at disaster sites, it may be necessary to consider a simpler format. Also, in combining telemedicine and measurement of mental health, the digital privacy of patients must be considered [19]. Second, although the mind–body modalities introduced in our manual were provided through YouTube videos, YouTube videos have one-way characteristics. Based on our clinical experience of mindfulness meditation, we believe the application of mind–body modalities, including mindfulness meditation, is more effective in an interactive communication environment. Specifically, a sufficient feedback process is required following each practice, and this may be achieved by using programs such as Zoom Meetings. In the digital interactive communication environment, such as via Zoom Technology, it is possible to consider the construction of an online community-based meditation practice that may contribute to improving public mental health via web-based social interaction. Third, although mindfulness meditation is a popular mind–body modality that is widely accepted, not only in Eastern cultures, but also in Western cultures, the cultural, ethnic, and religious/spiritual characteristics of patients should still be considered and respected. For example, in countries like China, Tai Chi or Qigong may be a more familiar movement to cultivate mindfulness [20], while in the United States, although not considered in our manual, spiritual meditation or mantra meditation may be good options [21]. Fourth, mental health telemedicine interventions for front-line healthcare providers also need to be developed. Our manual has been established for the general public or for infected patients, but today COVID-19 poses a serious risk for first-line medical staffs’ mental health [22]. Therefore, a revised manual may include strategies to improve mental health and relieve psychological stress for medical staff. Fifth, more specific scripts are needed for each stressful situation. For example, individual modalities could be developed according to events that may be applied to specific cases, such as family discord, social conflicts, helplessness, and despair. Finally, although our manual is limited, mindful movements such as yoga, Tai Chi, and Qigong may be useful strategies for mental health interventions through YouTube videos or software like Zoom Meetings. Importantly, because these movements transcend language barriers, they are likely to be helpful to foreigners residing in Korea, as well as other citizens residing outside Korea.

In view of these above limitations, future studies on mental health management using telemedicine for COVID-19 might consider the following issues. Researchers should consider adopting a validated and simple form of mental health assessment tool for the initial and follow-up assessments of individuals. A comprehensive review by Beidas et al. (2015) is helpful in developing the evaluation strategy [8]. Also, mental health interventions based on smartphone applications are increasing today, and some of these applications evaluate the emotional psychopathology of users through assessment tools such as the 9-item Patient Health Questionnaire or 7-item Generalized Anxiety Disorder scale [1]. Therefore, if the policy makers and information technology (IT) experts can consider and authorize the linkage of information between telemedicine and existing mental health applications, mental health assessment in the field of telemedicine is likely to improve. For the use of mind–body modalities in telemedicine, bidirectional communication between the individual and practitioner could be emphasized. YouTube videos or communication software, such as Zoom may facilitate bidirectional communication; such recent advances in online video technologies have increased the potential utilization of mindful movements such as yoga, Tai Chi, and Qigong.

## 4. Conclusions

The fear and social distancing caused by COVID-19 emphasized the importance of recognizing the mental health of all individuals. Here, mindfulness is a promising intervention that may be combined with telemedicine. Many attempts, such as telephone-adapted mindfulness-based stress reduction [23] and mHealth mindfulness intervention [24] have already been made. In this short paper, we introduced the “KM doctor’s mental health instruction manual in the telemedicine for COVID-19” as a pilot manual used by the patients attending the COVID-19 telemedicine center of KM in Korea. In this manual, a mindfulness-based intervention was introduced and may play an important role in assisting individuals faced with a pandemic or other emergency-situations. Based on our experience, we propose health authorities in other countries consider the establishment of telemedicine-based mental health management strategies and further share their experiences and potential research. For mental health care in the upcoming “new-normal” era, mindfulness-based interventions are promising mind–body modalities.

## Figures and Tables

**Figure 1 ijerph-17-04477-f001:**
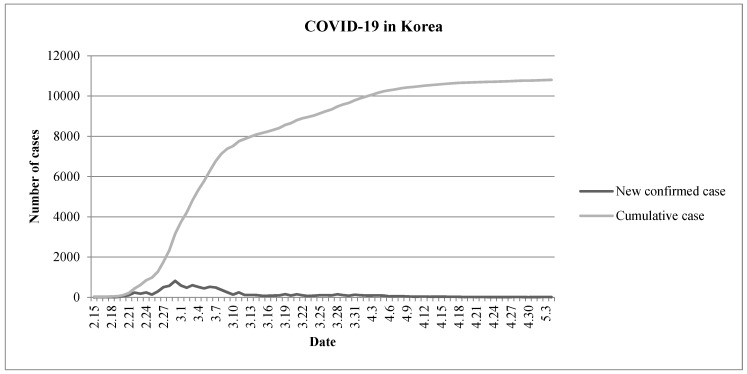
Changes in the number of COVID-19 patients in Korea (15 February–4 May 2020).

**Figure 2 ijerph-17-04477-f002:**
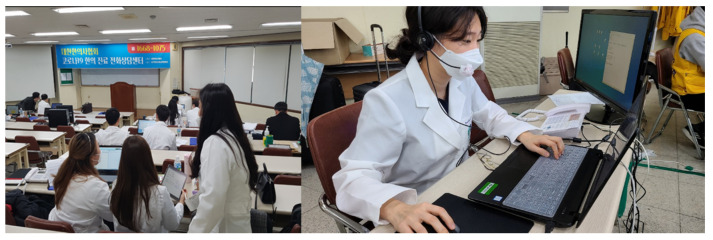
The COVID-19 telemedicine center of Korean medicine (KM). Note. The center has been treating more than 20% of confirmed COVID-19 patients in Korea with telemedicine since 9 March.

**Table 1 ijerph-17-04477-t001:** History taking: 3-step approach.

Steps	Questionnaire	
Step 1	“Do you feel distressed or need psychological support for symptoms such as overstrain, dyspepsia, and insomnia?”	
Step 2	“If so, can you quantify it? Please express it as a number between 0 and 10. Set the most severe level as 10 and answer 0 when there are no symptoms at all.”	
Step 3	Overstrain	“Do you think that you have been more nervous in your daily life than necessary?”
	Fear	“Are you struggling with fear or fear of the coronavirus?”
	Anxiety	“Do you have a lot of worries or thoughts that constantly make you feel restless and anxious?”
	Lethargy/depression	“Do you feel depressed without having fun, or are you feeling lethargy?”
	Insomnia	“Are you suffering because you haven’t been sleeping well these days? If you don’t sleep easily, wake up often in the middle night, or wake up too early in the morning, making you feel tired throughout the day, that means you are not sleeping well.”
	Dyspepsia	“Is it uncomfortable when you eat food these days? Are you reluctant to eat with reduced appetite or indigestion?”
	Pain	“Do you have new pain whenever you feel bad? Or do you feel more unpleasant pain in the areas where you felt pain before? ”
	Anger/irritability	“Have you easily become angry or annoyed these days?”

**Table 2 ijerph-17-04477-t002:** Guidance for symptom management.

Symptoms	Mechanism	Recommended Coping Strategy
Overstrain	Tension is a natural phenomenon of the human body to survive in the stress response theory (fight or flight reaction). The constant rumination creates overstrain, even in non-existent events.	Notice the thoughts that make you nervous. Imagine the thought of relaxing yourself repeatedly, and if this is difficult, focus on the physical stimulus to relax your body (e.g., half body bath, listening to music, walking, etc.)
Fear	The reaction of fear appears when the stress response to protect our body from danger is extremely severe. Fear is easily learned, and when a circuit of thought is formed for the fearful situation, the emotion can be reproduced whenever the situation appears or a related thought arises. If you experience panic, you may develop a fear (expected anxiety) that the symptoms will occur again. Once panic is experienced, expectation anxiety may follow.	In case of an excessive fear reaction, prevent hyperventilation and induce parasympathetic activity through exhalation-oriented deep breathing. Repeatedly reaffirm the sense of security that can cope with fear. If you are relaxed and calm enough, try exposing yourself to the usual stimuli that triggered your fear response.
Anxiety	Acutely, anxiety appears with the reaction of fear. Anxiety is closely related to the thought process. When a person falls into the thought that triggers anxiety, more and more thoughts are continually generated, and in this process, anxiety can be strengthened.	Notice that you are anxious. This is the first step in laying down the thoughts that cause anxiety. Instead of turning your attention to other thoughts, it is helpful to focus on your body sensations instead.
Lethargy/depression	Reactive depression can occur when an acute emotional reaction occurs, but the situation persists unchanged, and the mental energy that an individual can consume has reached a limit. In addition, individual vulnerability is a risk factor that easily causes lethargy and depression. Decreasing physical activity exacerbates depression.	Find out your own new rhythm in a small space called “home.” Discover and expand positive energies like charity, empathy, loving-kindness, and mercy that exist in your mind.
Insomnia	Depression, anxiety, and decreased physical activity can cause insomnia. Insomnia has a close relationship with cognitive factors, and if insomnia causes anxiety, this anxiety can exacerbate insomnia again. Bad sleep hygiene and sleep habits can also cause insomnia.	The solution to insomnia is based on the recovery of biorhythm. That is, a rhythm that is sufficiently active during the day and rests during the night should be restored. In case of worries, anxiety, and tension that persists insomnia, apply methods of relaxing the body and attempt mental distraction methods. Please observe good sleep hygiene. If you are taking sleeping pills, you need guidance and management on how to take the correct sleeping pills.
Dyspepsia	Depression can reduce appetite. If a person eats only similar foods in a limited space, and the number of people who can eat together is limited, depression and loss of appetite become more severe. If the sympathetic activity continues to be elevated, the movement of the digestive system is not smooth. This leads to a decrease in gastric motility, causing some symptoms, including dyspepsia.	Imagine the memory of eating something delicious before the current eating situation and promote your appetite. Eat meals regularly, and after eating, help the gastrointestinal tract to digest enough through physical activities, such as walking lightly. It is better to eat even a small amount of fun and delicious food, rather than eating it excessively and vigorously.
Pain	Local pain can be caused primarily by muscle tension in the area. Body pain is closely related to cognitive factors. Worries about the pain, psychological tension, anxiety, and depression can amplify the pain condition.	It is important to know that thoughts, feelings, and pain are closely related. Try some work or activity that reminds you of thoughts other than pain or that may make you forget about pain. Observe the pain from the point of view that it is not just a bad thing or an unpleasant event, but simply a signal or sensation from your body.
Anger/irritability	Anger emotion suggests the resistance to the irrational and absurd reality from the thought that you have been harmed by the current situation. Anger induces a state of tension in the short term, so it instantly boosts the body’s metabolism and activates the immune system. Chronic anger and tension, however, can depress the immune system and lead to depression, lethargy, and some somatic symptoms.	Understand that current anger is a natural reaction to the situation. Find other activities or ways to express this intense energy. Some relaxation modalities may be appropriate methods. It is important to be an objective third-party observer of this situation that is damaging to me.

**Table 3 ijerph-17-04477-t003:** Care algorithm of the manual.

Symptom		Education	Basic Modalities ^1^	Individual Modalities
Persistent symptoms	Overstrain	Expressing empathy for the symptoms, and educating on the mechanisms of each symptom	Simple breathing Mindful breathing Walking meditation	Progressive muscle relaxation Autogenic training
	Fear			Breath-counting meditation
	Anxiety			Sitting meditation
	Lethargy/depression			Loving-kindness meditation Eating meditation
Episodic symptoms	Physical symptoms (including insomnia, dyspepsia, and pain condition)			Body scan (for insomnia) Eating meditation (for dyspepsia) 15 min meditation with *qi* (for pain condition)
	Anger/irritability			Sitting meditation

**^1^** In the case of “basic modalities,” it is recommended the modalities be carried out regularly, such as once in the morning and evening, whenever possible. In the case of “individual modalities,” it is recommended to perform the suggested modalities when symptoms occur.

**Table 4 ijerph-17-04477-t004:** Mind–body modalities for symptom management.

Mind–Body Modalities	Description
Simple breathing	Repeat your inspiration and exhalation to find your original rhythm. Find the most stable, comfortable, and balanced one. Breathe and feel safe and comfortable.
Mindful breathing	Observe your breath. Let us observe the inspiration and exhalation. Try to feel cool air coming into your body and turbid air coming out of your body. Breathing confirms that your body is clear and healthy. Try to feel that your body is clear and healthy throughout this breathing exercise.
Walking meditation	Step on the ground and make sure it is stable and firm. While walking slowly, check that it is stable-unstable-stable again. Walk to your own rhythm and find yourself comfortable and balanced. Even in a small space, you can see the vitality of movement.
Progressive muscle relaxation	Divide the body parts and try to repeat the local tension and relaxation. Tension your muscles while you breathe in and relax your muscles while you exhale. This process begins with your hands and spreads to each part of the body. Throughout this process, make sure that your body is sufficiently relaxed.
Autogenic training	Try to create the most stable and relaxed state. Notice that both hands are warm. Notice that both hands are heavy. Notice that your heart beats regularly. Notice that your breathing is comfortable. Notice that your lower abdomen is warm. Notice that the forehead is cool.
Breath-counting meditation	Breathe with the numbers to focus more on your breathing. Each time you breathe in and out, count backwards starting with 10 to 1. Focus only on breathing and numbers, and if you have other thoughts, try to focus on the breathing again.
Sitting meditation	As you breathe comfortably, notice your body sensations, thoughts, and emotions. If a disturbing thought or emotion occurs, just observe it with tranquility. Observe how it changes. It is important to take a non-judgmental attitude, rather than resisting or interpreting its meaning.
Loving-kindness meditation	Let us check the warm heart we originally had. Think about the sadness that a mother feels when seeing a sick child, or the wish that the child will be cured. Let’s extend that warmth to me, to my family, to my friends, and to my health care provider.
Eating meditation	Try to be mindful when eating. Do not rush food automatically, see it with your eyes, take it with the nose, taste it with the tongue, chew it with the teeth, swallow it with the throat, pass it through the esophagus, fill the stomach, and eventually feel satisfied. Also, imagine that the energy generated in this process is supplied to the whole body.
Body scan	Observe your body closely. Identify the sensations, feelings, or pains felt in each area and accept it as it is. Just accept it while looking. In the process of scanning the whole body, make sure that your body and mind are gradually relaxed.
15 min meditation with *qi* (energy)	Feel the warm energy in your palms. Use that warm energy to deliver it to places where your body is uncomfortable or where you are in pain. Make sure that the warmth is relaxing your symptoms and pain. When you are done, put your hands on top of your belly and deliver the warm energy to your body.

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
