# Peer review of "Using Mind–Body Modalities via Telemedicine during the COVID-19 Crisis: Cases in the Republic of Korea"

_ijerph, 2020, doi:10.3390/ijerph17124477_

Round 1

Reviewer 1 Report

In the new version presented, the authors improved the manuscript using all the suggestions I had previously provided.
As the paper already had enough merits to be appreciated, with the present improvement, I think it can be published.

Reviewer 2 Report

Thank you for making the suggested changes - this is a very interesting paper.

This manuscript is a resubmission of an earlier submission. The following is a list of the peer review reports and author responses from that submission.

Round 1

Reviewer 1 Report

The manuscript describes a success case about a distance education initiative on how traditional Korean medicine resources can help promote balance in people with impaired mental health due to the pandemic.
It is a well-organized material with practical application that can encourage similar initiatives in other countries, even those that officially adopt only Western medicine, with the necessary adaptations.
The following suggestions are for improving the academic format of the paper. I ask the authors to please:

- In ABSTRACT, in Background, more emphasis on the pandemic effects on mental health and how complementary therapies can be potentially useful. Both information are the motivation of the initiative.
- Line 79: Explain better the telemedicine center beyond the university affiliation of coordinators. Is it a government or private initiative? Did it just appear now, or was it a branch from some previous initiative?
- Inform in what it was inspired (previous cases of telemedicine for mental health during catastrophe, either published or not); if none were found, quote this in the description of its creation.
- After the list of Limitations of the initiative, inform what future studies (whether by these authors or others) should focus (follow-up of beneficiaries on the maintenance of results, etc.).

Author Response

Authors’ Response to the Reviewers’ Comments

Journal: International Journal of Environmental Research and Public Health

Manuscript number: ijerph-826298

Title: Telemedicine Using Mind–Body Modalities during the COVID-19 Crisis: Cases in the Republic of Korea

  • Response to Comments from Reviewer 1

Overall comment:

The manuscript describes a success case about a distance education initiative on how traditional Korean medicine resources can help promote balance in people with impaired mental health due to the pandemic.

It is a well-organized material with practical application that can encourage similar initiatives in other countries, even those that officially adopt only Western medicine, with the necessary adaptations.

The following suggestions are for improving the academic format of the paper. I ask the authors to please:

Response:

We greatly appreciate the reviewer’s comments and have carefully reviewed the manuscript. We believe that the changes made to the manuscript based on the reviewer's comments has made our manuscript more valuable.

Comment 1:

- In ABSTRACT, in Background, more emphasis on the pandemic effects on mental health and how complementary therapies can be potentially useful. Both information are the motivation of the initiative.

Response 1:           

Based on the reviewer's comments, we have added details on how this pandemic affects mental health and what the potential benefits of complementary therapies are in the Abstract and in the Background sections of the manuscript.

(please see page 1 of the revised manuscript and the title page, marked in red)

Comment 2:

- Line 79: Explain better the telemedicine center beyond the university affiliation of coordinators. Is it a government or private initiative? Did it just appear now, or was it a branch from some previous initiative?  

Response 2:           

The Association of Korean Medicine (AKOM) established the COVID-19 telemedicine center of Korean medicine (KM telemedicine center) at Daegu Korean Medicine Hospital, Daegu, Korea. AKOM is a representative organization of all KM doctors in Korea. After the outbreak in Daegu and Gyeongbuk was stabilized, AKOM also established an additional telemedicine center in Seoul. In order to better manage the mental health of victims and survivors of COVID-19, the center requested Prof. Jong Woo Kim to develop a mental health instruction manual using mind-body interventions, including mindfulness meditation techniques. We have added these details in the Background section to make it more clear.

 (please see pages 2-3 of the revised manuscript and the title page, marked in red)

Comment 3:

- Inform in what it was inspired (previous cases of telemedicine for mental health during catastrophe, either published or not); if none were found, quote this in the description of its creation.  

Response 3:           

As is the case in other countries, this COVID-19 pandemic caused an unprecedented disaster in Korea, and there were no previously published cases of the use of telemedicine for mental health during such catastrophes. In addition, due to the urgency of manually developing the instruction manual, it was difficult to conduct a comprehensive systematic literature review methodology. Instead, the development team developed this manual on the basis of empirical evidences obtained from narrative review methodology, as well as, their experience in clinical settings. We have added these statements under section 2.1 Development of the manual.

 (please see page 3 of the revised manuscript and the title page, marked in red)

Comment 4:

- After the list of Limitations of the initiative, inform what future studies (whether by these authors or others) should focus (follow-up of beneficiaries on the maintenance of results, etc.).  

Response 4:           

We have added statements under section 3. Limitations of the manual and further suggestions to emphasize and clarify what we should focus on in our future studies.

 (please see page 9 of the revised manuscript and the title page, marked in red)

Reviewer 2 Report

Thank you for this very interesting paper which reports Telemedicine Using Mind–Body Modalities during the COVID-19 Crisis. I am very interested in this topic and I read about your manual with interest. This is not a research study as such but rather the description of a manual developed to support staff to help patients during telemedicine consultations. It is not clear how this manual was put together or what evidence was used to inform this? Was there a literature review completed for example. Or did an Advisory Group inform the development? Is this an evidence  based project. I attach the paper with further comments.

Author Response

Authors’ Response to the Reviewers’ Comments

Journal: International Journal of Environmental Research and Public Health

Manuscript number: ijerph-826298

Title: Telemedicine Using Mind–Body Modalities during the COVID-19 Crisis: Cases in the Republic of Korea

  • Response to Comments from Reviewer 2

Overall comment:

Thank you for this very interesting paper which reports Telemedicine Using Mind–Body Modalities during the COVID-19 Crisis. I am very interested in this topic and I read about your manual with interest. This is not a research study as such but rather the description of a manual developed to support staff to help patients during telemedicine consultations.

Response:

We greatly appreciate the reviewer’s comments and have carefully reviewed the manuscript. We believe that the changes made to the manuscript based on the reviewer's comments will make our manuscript more valuable.

Comment 1:

It is not clear how this manual was put together or what evidence was used to inform this? Was there a literature review completed for example. Or did an Advisory Group inform the development? Is this an evidence based project. I attach the paper with further comments.

Response 1:           

We acknowledge that the description of the manual development process in the original manuscript was poor. Following the reviewer's comments, we added a detailed description on the development of this manual.

In addition, all of the valuable comments provided by the reviewer on the attached material were reflected in the revised manuscript. In particular, we tried to replace the existing emotional language with a more subjective and focused language.

For the content of FAQ, however, we suggest that it is appropriate to present this section in the main text of the manuscript rather than as an appendix, as it is an important part of the telemedicine consultation process. These are the recommendations to be followed by KM doctors based on the frequent answers given by the  patients.

(please see page 3 of the revised manuscript and the title page, marked in red)

Round 2

Reviewer 2 Report

Thank you for making the suggested revisions and offering explanations.